# Vocational rehabilitation for people with multiple sclerosis in the national health service of the United Kingdom: A realist evaluation

**Blanca De Dios Perez**[1,2*], **Vicky Booth**[1,3], **Roshan das Nair**[4,5,6], **Nikos Evangelou**[4], **Juliet Hassard**[7], **Helen L. Ford**[8,9], **Ian Newsome**[10], **Kate Radford**[1,2]

**1** University of Nottingham (Centre for Rehabilitation and Ageing Research, School of Medicine), Nottingham, United Kingdom, **2** NIHR Nottingham Biomedical Research Centre, Nottingham, United Kingdom, **3** Nottingham University Hospitals NHS Trust, Nottingham, United Kingdom, **4** University of Nottingham (Mental Health & Clinical Neurosciences, School of Medicine), Nottingham, United Kingdom, **5** Nottinghamshire Healthcare Trust (Institute of Mental Health), Nottingham, United Kingdom, **6** SINTEF (Health Division), Trondheim, Norway, **7** Queen's University Belfast, Belfast, United Kingdom, **8** Leeds Teaching Hospital NHS Trust, Leeds, United Kingdom, **9** University of Leeds; Leeds, United Kingdom, **10** Lay co-author, York, United Kingdom

\* blanca.dediosperez@nottingham.ac.uk

## Abstract

### Background

There is limited evidence about how vocational rehabilitation (VR) for people with multiple sclerosis (MS) can be delivered through the United Kingdom's (UK) National Health Service (NHS) and how it works.

### Aim

To understand the mechanisms and context for implementing a VR intervention for people with MS in the NHS and develop an explanatory programme theory.

### Methods

A realist evaluation, including a review of evidence followed by semi-structured interviews. A realist review about VR for people with MS in the NHS was conducted on six electronic databases (PubMed, MEDLINE, PsychINFO, Web of Science, CINAHL, and EMBASE) with secondary purposive searches. Included studies were assessed for relevance and rigour. Semi-structured interviews with people with MS, employers, and healthcare professionals, were conducted remotely. Data were extracted, analysed, and synthesised to refine the programme theory and produce a logic model.

### Results

Data from 13 studies, and 19 interviews (10 people with MS, five employers, and four healthcare professionals) contributed to producing the programme theory. The resulting

**Data availability statement:** The data extraction forms from the Realist Evaluation have been included in a Data Repository and can be accessed through the following link: https://rdmc.nottingham.ac.uk/handle/internal/11597 There are restrictions related to the consent provided by the study participants for access to the data (interview transcripts) they provided for the study. The University of Nottingham (UoN), the Health Research Authority (HRA) for the United Kingdom (UK), and the London Stanmore Research Ethics Committee (REC) approved the study. There are some ethical restrictions on sharing the research participant data due to the presence of potentially identifiable and sensitive patient information about personal and work experiences, barriers to accessing healthcare, and instances of discrimination at work. Participants only consented to quotes from their interviews being included in the final study but not the full transcript. The University of Nottingham is legally responsible for data security, and the Chief Investigator (Prof. Kate Radford) is the Data Custodian managing the access to the data. Reasonable request for access to the data reported in this study may be sent to the Faculty of Medicine & Health Sciences Research Ethics Committee of the University of Nottingham MZ-FMHS-ResearchEthics@exmail.nottingham.ac.uk (study reference: FMHS 477-0322).

**Funding:** The UK Multiple Sclerosis Society funded this research through a research project grant (award:140). The funders had no role in study design, data collection and analysis, decision to publish, or preparation of the manuscript

**Competing interests:** The authors (VB, NE, JS, HF, IN, and KR) have declared that no competing interests exist. I have read the journal's policy and the authors of this manuscript have the following competing interests: Author BDP has received funding from the Neurology Academy (speakers bureau) to deliver a lecture on vocational rehabilitation for people with MS. Author RdN has received funding (speakers bureau) from Novartis, Biogen, and Merck for delivering lectures on psychological aspects of MS and cognitive screening and rehabilitation in MS.

programme theory explains the implementation of VR in the NHS for MS populations, uncovering the complex interplay between the healthcare and employment sectors to influence health and employment outcomes. VR programmes that offer timely support, tailored to the needs of the person with MS, and that support and empower the employee beyond the healthcare context are most likely associated with improved employment outcomes, for example, job retention.

## Conclusion

Embedding VR support within the NHS requires substantial cultural and organisational change (e.g., increased staff numbers, training, and awareness about the benefits of work). This study emphasises the need to routinely identify people with MS at risk of job loss and follow a collaborative approach to address employment issues. This realist evaluation provides insight on how to improve the quality of care available to people with MS.

## Introduction

Work can be a part of a person's identity, leading to financial independence, providing a purpose in life, improving self-esteem and reducing healthcare costs [1,2]. Yet, many people with health conditions, such as multiple sclerosis (MS) experience difficulties at work and tend to leave the workforce prematurely [3].

Vocational rehabilitation (VR) is "a process whereby people affected by illness or disability are unable to remain, return or find new employment" [4]. It is recommended that VR interventions for people with long-term neurological conditions receive support early after diagnosis, tailored to their needs, allowing for prompt follow-up of new issues and open access (i.e., people can re-access services over time) [5]. In the UK, VR interventions are provided in various settings such as the private sector, Department for Work and Pension (DWP), or healthcare settings [6]. In fact, employment is an outcome of health interventions in the UK National Health Service (NHS) [7].

VR has been a priority in the UK for several years, highlighting the benefits these interventions could have on the employment of people with illness and disabilities [8]. More recently, in the UK Government's 2023 Autumn statement, there was a push for the need to support people with illness and disabilities to return or remain employed [9], driven in part by the increase in people off work due to sickness absence since the COVID-19 pandemic reaching 2.5 million people in December 2022 [10].

Whilst VR is a promising intervention to improve the employment rates of people with disabilities, VR interventions are complex. They are characterised by having multiple intervention components (e.g., symptom management, changing employer attitudes), are highly individualised to the employee and their role, and can lead to changes in multiple outcomes (e.g., confidence levels, productivity, employment rates) for different stakeholders [11]. The contexts where these interventions are delivered are also varied. Thus, when measuring the effectiveness of these interventions, it is not sufficient to understand whether VR works (or not), but under what circumstances (*context*).

MS is a good example of a healthcare condition where patients may benefit from VR. MS is the most common chronic neurological condition affecting young adults, with an average age of diagnosis between 20–40 [12]. The physical, psychological, and cognitive problems that people with MS experience can create barriers to job retention [13,14]. The UK National Institute for Health and Care Excellence (NICE) guidance for treating adults with MS states

that when a person is diagnosed with MS, healthcare professionals should provide information regarding work to support them [15]. Despite the possible benefits of VR for this patient group and the existence of national healthcare guidance for MS-related work, VR is still not readily available for people with MS in NHS settings. This is partly due to the complexity of delivering VR.

Using realist methodology to understand complex interventions such as VR is increasingly common because it explains how the context can influence different stakeholders' behaviours and how these behaviour changes lead to outcomes [16,17]. For example, realist approaches were used to understand a model of early intervention VR for people with acquired brain injury in the healthcare system of New Zealand [18]. Therefore, this methodology may enable us to understand how VR interventions for people with MS work. This is important because the evidence on the effectiveness of VR interventions for people with MS is inconclusive, in part because there is a lack of randomised controlled trials (RCTs) and the multiple VR outcomes that have been studied (i.e., return to work, job retention), which hamper the synthesis of the evidence [19].

Therefore, this study aimed to understand what VR interventions for people with MS delivered in the NHS of the UK work, under what circumstances, and why. The realist review will answer the following questions: (1) What are the important contexts within the NHS that determine whether the different mechanisms within a VR intervention for employed people with MS produce the intended outcomes?; (2) What are the mechanisms, acting at an individual and organisational level, by which VR interventions for employed people with MS produce the intended outcomes (e.g., job retention)?; (3) What are the possible outcomes of a VR intervention for employed people with MS?

## Materials and methods

We conducted a realist evaluation, drawing on the reporting standards for realist evaluations (RAMESES II) [20] (S1 File). We included a realist review, stakeholder interviews, and produced a programme theory using a logic model format. The protocol for the realist review was registered in PROSPERO (registration: CRD42022315542) [21].

### Development of initial rough programme theory

Previous research conducted by the authors about VR and people with MS and discussion with the authors informed the initial rough programme theory [22–24] (Table 1). It was developed to account for theories at micro, meso- and macro levels (individual, interpersonal, and cultural) [25–27], and to give an initial rough theory upon which to base the literature review.

The initial rough programme theory was developed based on context-mechanisms-outcome (CMO) configurations about how the intervention would work. The following definitions were used to support the analysis: the context refers to the NHS setting or stage when the intervention will be delivered; the mechanisms refer to how the VR intervention works and what it triggered in the person with MS (e.g., changes in feelings or thoughts); the outcomes refer to the consequences (expected and unexpected) of the mechanisms in each context.

### Evaluation data collection

**Realist review.** The evaluation started by scoping the literature on VR interventions for people with MS delivered as part of NHS services. Systematic searches in six electronic databases (PubMed, MEDLINE, EMBASE, Web of Science, CINAHL, and PsyINFO) were conducted from inception until 3rd November 2023. Secondary searches for additional literature were conducted in Google Scholar, Ethos, British Library, and

**Table 1. Initial rough programme theory.**

| Initial Programme Theory | Supporting evidence | Theory Level |
|---|---|---|
| **Context** | | |
| **Employment is discussed as part of the usual care** of people with MS because healthcare professionals acknowledge the relevance of employment. | Identify by authors | Macro |
| **Mechanism** | | |
| VR for people with MS is provided soon after diagnosis [**early intervention**]. | [28–30] | Macro |
| The intervention is **individually tailored** to the needs of the person with MS. | [5] | Micro |
| The employer is included in the intervention [**Employer engagement**] | [30,31] | Meso |
| The VR therapists build a relationship of **trust** with the person with MS and the employer (if included) to facilitate the **collaborating** towards removing barriers to job retention. | Identified by authors | Micro |
| **Outcome** | | |
| The person with MS receives **reasonable accommodations** to manage the impact of the disability at work. | [5,23,32] | Macro |
| The employer gains an understanding of MS and how to support their employee at work [**improved likelihood of job retention**]. | Identified by authors | Macro |

The words in **bold** are the key components of the initial rough programme theory. Theory Level: Micro = individual (person with MS, employer, healthcare professional); Messo = team (implementer team, or person with MS + care team, employers, etc.); Macro = organisational level (hospital, clinic, workplace).

Clinicaltrials.gov.uk. These searches included a combination of the terms VR, MS, and NHS. The primary and secondary search strategies are included in S2 File.

The references of relevant manuscripts and systematic reviews were reviewed to identify further evidence following the "snowball" approach. Realist reviews do not exclude evidence based on their study design [33]; therefore, no study design filters were used in the searches.

We developed a screening, data extraction and appraisal tool that were piloted and refined to aid the review process. The screening tool included questions regarding the topic, methodology, and content to aid in identifying evidence for the review. The data extraction tool (S3 File) included information on the relevance of the document, a table to extract CMO configurations, judgement about how the programme theory was refined or refuted based on the data extracted, and a free text section for the reviewer's comments.

A tool for assessing rigour and relevance was developed (S4 File). Pawson et al. [34] TAPUPAS criteria were used to assess issues regarding rigour, and relevance was assessed based on how many aspects of the programme theory were covered by the document.

Following a realist approach, we sought patterns of factors that affected intervention outcomes and extracted data following CMO configurations. If a component of the CMO was missing, data dyads (e.g., C-O; C-M) were extracted.

## Interviews

Semi-structured interviews with stakeholders (people with MS, employers, and healthcare providers) were conducted following a realist approach to discuss the findings from the literature and refine the programme theories [35]. Participants were recruited from 26 May 2022 to 21 February 2023. Participants with MS were recruited if they had (1) a neurologist-confirmed diagnosis of MS, (2) were currently employed, and (3) had previously requested support with employment from the NHS. Employers were included if they had experience managing an employee with MS at work. Healthcare professionals were recruited if they had previous experience delivering vocational rehabilitation to people with MS or an interest in employment. Stakeholders were presented with CMO configurations identified in the realist review and subsequently included within the initial rough programme theory [25,36].

Interviews were conducted following a realist approach and using a topic guide (S4 File). The interviews explored the engagement of people with MS with NHS services, experiences reporting employment issues at healthcare appointments, and approaches to integrating employment and healthcare services. Data were analysed following a similar approach as the realist review by extracting CMO configurations.

Ethical approval was obtained from the Faculty of Medicine & Health Sciences Research Ethics Committee (REC) at the University of Nottingham (reference: FMHS 477-0322) and NHS Ethical Approval from the Stanmore REC (reference: 22/PR/1030). The study was conducted in accordance with the Declaration of Helsinki. All participants completed a written consent form before data collection.

## Data analysis

One author (BDP) led the data analysis and reviewed the progress with a second author (VB). Data were coded based on CMO configurations following the aforementioned definitions for each component of the programme theory (CMOs).

## Realist review

BDP completed data extraction and coding for all studies, and data were entered into an Excel document. Two authors (BDP, KR) defined the main mechanisms of the initial programme theories to support the data extraction and coding process and developed a matrix to code the CMOs identified. Challenges associated with the data coding were discussed with a third author (VB).

The data extraction aimed to identify the elements that lead to the success or failure of VR interventions for people with MS in the NHS from a realist perspective. We extracted data regarding the intervention characteristics, information about the contexts involved in the delivery of the intervention, mechanisms, and outcomes (including descriptions about why the intervention worked or why it did not work).

## Interviews

Interviews were audio-recorded, transcribed verbatim, and systematically coded following a hybrid deductive and inductive approach. Data were analysed deductively according to CMO configurations following a realist evaluation approach [35,37], followed by an inductive data-led analysis to develop further our understanding of how VR could work in the NHS.

## Data synthesis

Data were synthesised following the steps described by Wong et al. [33]. The CMO configurations were organised according to the mechanisms. We looked for patterns of demi-regularities (context-outcome) across the literature to look for consistency patterns across the data and understand the underlying mechanisms of the VR intervention.

A series of "if-then-because" statements were developed iteratively to summarise the CMO configurations. This process allowed refining or refuting the original programme theories based on the data analysed.

We combine the data from the literature and the interview participants' personal experiences through an iterative process to explore the intervention's underlying mechanisms and refine the initial rough programme theory. While some mechanisms had clear patterns and meanings, others were identified through discussion with the research team, which had extensive expertise in MS and VR.

The results are presented following the mechanisms identified, and the programme theory was refined based on the findings. Production of the logic model was the final phase of the data synthesis.

### Patient and public involvement

A patient and public involvement (PPI) representative (IN, White British man) was involved in developing the study protocol, topic guide, and data analysis to improve the validity of the findings and ensure patients' perspectives are included in the research.

## Results

### Realist review

The searches identified 7,209 studies, of which 1,852 were duplicates (Fig 1). Of the 5,357 unique studies identified, 65 were eligible for full-text screening, and 13 studies were included in the review (Table 2). The reasons for excluding studies are presented in Fig 1.

There was a varied range of research designs and publications included in the review: six service evaluations [38–43], two guidelines [15,44], one qualitative study [45], one case study [46], one thesis reporting on a feasibility RCT [31], one survey study [47], and one protocol for a feasibility randomised study [48].

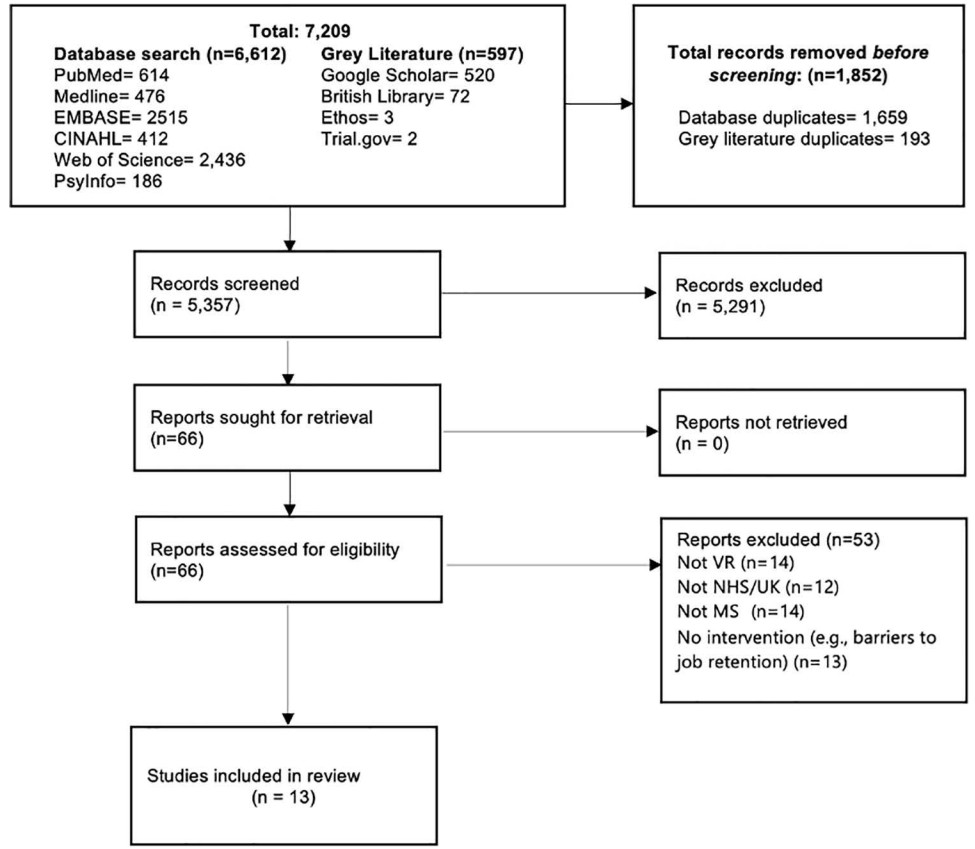

**Fig 1. Screening flowchart.**

**Table 2. Summary of studies identified.**

| Reference | Study design | Objectives | Sample | Setting | Main findings |
|---|---|---|---|---|---|
| Sweetland et al. (2014) | Case Study | To explore what is meant by early intervention in VR. | N = 2 People with MS | Outpatient long-term neurological conditions service. | Early job retention VR intervention for people with MS. Occupational Therapist (OT) lead intervention. Characterised by individually tailored intervention with up to six sessions on consecutive weeks lasting 1.5 hours. Components include education, support identifying and implementing reasonable adjustments, disclosure, psychological adjustments, support managing performance, and employer engagement. |
| Sweetland (2010) | Feasibility RCT | To develop and implement an OT lead VR job retention intervention to support people with MS to remain at work. | N = 27 | | |
| Jellie et al. (2014) | Qualitative Study | To explore the experiences of receiving a VR intervention. | N = 19 | | Five major themes related to the impact of the intervention on "understanding my symptoms and their management in the workplace", "removing my anxieties", "understanding and influencing my employer", "managing my loss of confidence" and "having professional support". The VR intervention was valued by people with MS who felt that after the intervention they had a greater understanding of disease related, work related and personal factors that impacted on their ability to work, and they were supported to manage these by a skilled professional. |
| Townsend (2008) | Survey | To explore the knowledge and experience of professionals supporting people with MS in work; identify current practice and training needs of professionals supporting people with MS in work | N = 70 OTs: 32 MS Specialist nurse: 26 MS specialist physiotherapist: 4 Disability employment adviser: 8 | – | Only 80% (n = 56) of participants reported on current support: -Information provision (93%, n = 52) -Support completing forms (37.5, n = 21) -Individual programmes of intervention or advice (23%, n = 13) -Group interventions (14.3%, n = 8) -Other [joint assessments (17.8%, n = 10), working with non-health social services (14.2%, n = 8)] |
| McGregor (2014) | Service Evaluation | To evaluate the impact of a vocational rehabilitation service pilot. | N = 303 People with cancer (73%), MS (10%) and IBD (17%) | NHS Greater Glasgow and Clyde | Tiered case management model (**Tier 1**: self-help, **Tier 2**: moderate support (information to manage health at work, positive message about work, signposting/referrals; **Tier 3**: specialist intensive support) 45% of clients were off work sick and looking to get back to work. The service was associated with improved quality of life (measured by EQOL5D (self-care domain), HADS (reduced anxiety and depression), and increasing the numbers of people in work and reduction in the numbers off sick. 92% of people who were in work when discharged were still in work. Some clients who were not at work when they were discharged had also returned to work so that the overall proportion of those in work at discharge has increased from 66% to 90%. |
| Bisiker & Millinchip (2007) | Service Evaluation | Retrospective audit to review the progress of the "Equal Pathways to Work" project. | N = 74 (56 men, 18 women) Brain injury (n = 8), head injury (n = 27), stroke (n = 23), MS (n = 5), Guillian-Barre syndrome (n = 2), other (n = 9) | Rehabilitation Unit West Park Hospital, Wolverhampton | Only 25 out of 74 people referred to the service maintained or obtained employment. This is partly because most people were in training at the time of the service evaluation. 40% of participants with MS returned to work. There was not an emerging pattern when comparing diagnosis and the success in returning to work. |
| Kirker et al. (1995) | Service Evaluation | To evaluate the workload and benefits of a new liaison nurse service for MS patients. | N = 136 (MS only) | Secondary Care (Medical Neurology Unit- Western General Hospital, Edinburgh) | Almost all newly diagnosed patients and any other patients who were having problems and who lived in or near Edinburgh were referred to the liaison nurse. She provided information about MS, encouragement and support to patient and family. Information on employment status was recorded. At referral she assessed needs and co-ordinated social services, welfare, OT, physiotherapy, wheelchair or driving assessment, etc. Feedback from 71 participants (87%). Better job prospects were reported by 7% of participants due to information about training schemes and working from home. |

*(Continued)*

**Table 2.** (Continued)

| Reference | Study design | Objectives | Sample | Setting | Main findings |
|---|---|---|---|---|---|
| Brewin & Hazell (2004) | Service Evaluation | Retrospective audit | N = 16 head injury (n = 5), stroke (n = 6), rheumatoid arthritis (n = 3), MS (n = 2). | Rehabilitation Unit West Park Hospital, Wolverhampton | Participants completed an assessment of needs, received a written OT report, and had a joint meeting between OT, client and DEA, to either plan a workplace visit or agree a plan of action. The progress was reviewed based on need and further recommendations were made. The audit showed that only 50% of participants (n = 8) received the OT report, 62.5 (n = 10) had a joint meeting with DEA and OT, and only 12.5% (n = 2) received a workplace visit. Only 50% (n = 8) of participants achieved successful employment by the time of the audit. This was evenly distributed across conditions. |
| Wade (2011) | Service Evaluation | National Audit NHS of services for people with MS | N = 704 *people asked about VR* | NHS England and Wales | Specialist VR is not available or provided to most people in most areas. Only 11% of 168 people unable to return to work and 15% of 265 people with problems at work had any VR support. People with MS report that GPs lack knowledge about MS and its problems. Overall, they have positive attitudes towards specialist MS nurses and therapists. People with MS also expect a decline in the NHS services available in the future. |
| Main & Haig (2006) | Service Evaluation | Audit VR outpatient service to determine demand and effectiveness. | N = 76 Brain injury (n = 31), cardiac (n = 16), cerebro-vascular accident (n = 16), neurological (including MS) (n = 7), chronic pain (n = 5), amputees/loco-motor (n = 1) | Outpatient OT service (Astley Ainslie Hospital) | Two-thirds of patients needed advice to RTW. Interventions varied from support developing new skills, to negotiating with employers or other agencies. There were successful vocational outcomes, with 65 participants keeping their jobs/placements while attending VR, and 46 of them returning to their previous vocational pursuit. Six participants were still receiving support at the time of the audit. The success was in part attributed to offering support soon after illness onset (*early intervention*). |
| Ford (2020) | Randomised study | Support job retention in people with MS | Target of 92 participants with MS | Leeds Teaching Hospitals NHS Trust | The study aims to provide acceptance and commitment therapy to improve self-efficacy which has been shown to be a significant factor for helping people with MS who want to work to stay in work. The support is received online, using a web-based app to work through content at the pace of the person with MS (i.e., self-help). |
| NICE (2022) | Guidelines | Offer recommendations for the management of adults with MS | – | NHS | At the time of diagnosis, people with MS should receive information on their legal rights including social care, employment rights and benefits. |
| NICE (2014) | Guidelines | Management of MS in primary and secondary care | – | NHS primary and secondary care | People with MS should have a comprehensive review of all aspects of their care at least once a year, assessing for example the need for VR support or rehabilitation. |

MS, multiple sclerosis; OT, occupational therapist; IBD, inflammatory bowel disease; DEA, disability employment adviser; NICE, National Institute for Health and Care Excellence; VR, vocational rehabilitation; RTW, return to work; RCT, Randomised Controlled Trial.

## Quality appraisal

All papers were quality appraised for rigour and relevance to the programme theory (S6 Table). The studies included had a high methodological credibility, but their relevance was limited because most studies provided limited information on the middle-range theories.

## Semi-structured interviews

The characteristics of the participants recruited for the interviews are presented in Table 3.

## Data synthesis

Following the data synthesis process, the 'VR in NHS for MS Programme Theory' was produced (Table 4). Within it are two contexts (diagnostic appointment and routine care),

**Table 3. Demographic, clinical, and employment characteristics of participants.**

| | MS (n = 10) | Employers (n = 5) | Healthcare Professionals (n = 4) |
|---|---|---|---|
| Age [mean (SD)] | 49.3 (7.41) | | |
| Women | 7 (70%) | 3 (60%) | 4 (100%) |
| Men | 3 (30%) | 2 (40%) | 0 |
| Ethnicity* | | | |
| White British | 9 (90%) | 5 (100%) | 2 (50%) |
| Other white backgrounds | 1 (10%) | 0 | 0 |
| Indian/British Indian | 0 | 0 | 1 (25%) |
| Mixed/multiple ethnic backgrounds | 0 | 0 | 1 (25%) |
| Education | | | |
| A-Levels | 2 (20%) | 0 | 0 |
| GCSE | 2 (20%) | 1 (20%) | 0 |
| Degree | 1 (10%) | 2 (40%) | 2 (50%) |
| Postgraduate | 5 (50%) | 2 (40%) | 2 (50%) |
| Relationship Status | | | |
| Single | 2 (20%) | | |
| In a relationship | 8 (80%) | | |
| MS Characteristics | | | |
| Years living with MS | 8.25 (7.8) | | |
| RRMS | 6 (60%) | | |
| SPMS | 2 (20%) | | |
| PPMS | 2 (20%) | | |
| Employment characteristics | | | |
| Unemployed | 1 (10%) | 0 | 0 |
| Employed | 9 (90%) | 5 (100%) | 4 (100%) |
| *full-time* | 4 (40%) | 3 (60%) | 4 (100%) |
| *part-time* | 5 (50%) | 2 (40%) | 0 |
| Job Category | | | |
| Level 4 (Professional and managerial) | 4 (44.4%) | 3 (60%) | 3 (75%) |
| Level 3 (Associated professional and technical/ skilled trade) | 4 (44.4%) | 2 (40%) | 1 (25%) |
| Level 2 (Administrative, caring, leisure, sales, customer service, process, plant and machinery operatives) | 1 (11.1%) | 0 | 0 |
| Level 1 (Elementary occupation) | 0 | 0 | 0 |
| Employer Type | | | |
| Private | 5 (55.5%) | 1 (20%) | 1 (25%) |
| Public | 4 (44.4%) | 2 (40%) | 3 (75%) |
| Voluntary | 0 | 2 (40%) | 0 |
| Organisation size | | | |
| Large (>250 employees) | 8 (88.8%) | 3 (60%) | 3 (75%) |
| Medium (50-249) | 0 | 0 | 0 |
| Small (10-49) | 0 | 2 (40%) | 0 |
| Micro (<10) | 1 (11.1%) | 0 | 1 (25%) |
| Employment Sector | | | |
| Healthcare | 3 (33.3%) | 1 (20%) | 4 (100%) |
| Financial Services | 3 (33.3%) | 1 (20%) | 0 |
| Transport | 1 (11.1%) | 0 | 0 |
| Government | 1 (11.1%) | 0 | 0 |

*(Continued)*

**Table 3.** (Continued)

| | MS (n = 10) | Employers (n = 5) | Healthcare Professionals (n = 4) |
|---|---|---|---|
| Insurance Sector | 1 (11.1%) | 0 | 0 |
| Education | 0 | 1 (20%) | 0 |
| Tertiary Sector | 0 | 2 (40%) | 0 |

Organisation size obtained from UK Government guidelines; Job category obtained from UK Standard Occupational Classification (28).

MS, multiple sclerosis; RRMS, relapsing-remitting MS; SPMS, secondary progressive MS; PPMS, primary progressive MS.

*We use UK Census categories to describe ethnicity.

five mechanisms (early intervention, individually tailored, crossing health and employment boundaries, coordinated effort, and empowerment) and five outcomes (fostering hope, improving workability, improving workplace relationships, receiving reasonable adjustments, improving health outcomes, and job retention). Table 4 summarises the CMOs and supporting data within the programme theory.

## Early intervention

If people with MS are asked about their employment at the point of diagnosis, then healthcare professionals will be able to identify worries and anxieties about barriers to job retention, reduce the time of inactivity, and prevent the loss of confidence and self-esteem [38,46,47]. For those who have had a recent relapse, if they are provided with support with employment, then they will have an improved likelihood of returning to work [38,46]. Interview participants with MS reported that if they had known that there was a service to help them with employment issues in the future if needed, then they would have felt more hopeful and supported at the point of diagnosis.

If people with MS are not offered support with employment, then they will experience difficulties self-managing their condition at work (including learning how to manage fatigue, and cognition), are less likely to receive reasonable adjustments, and, by extension, are more likely to worry about their future at work and leave the workforce prematurely [39,45,47]. The timing of the information is essential, because if provided too soon after diagnosis, people with MS and their families may reject the support offered [44].

Once those needing VR are identified, they should be referred to specialist services using a referral system that allows multiple referral approaches (e.g., GPs referrals, self-referral, neurologist referral, etc) [31,43,45].

For early identification to be successful, interview participants suggested a need to highlight the importance of work to professionals working in the NHS. There is evidence suggesting that healthcare professionals may not be confident addressing the topic of "work" [47], and should be provided with additional training to understand the benefits of work, how to identify those in need of VR support, and improve their confidence to ask about work in routine appointments.

## Individually tailored

If healthcare professionals assess the person's employment needs, then they will be able to identify the main barriers to job retention and approaches to overcome these difficulties at work. Researchers recommended to conduct an initial assessment as the first stage of the intervention [31,38,40,41,43–46], allowing the intervention to be tailored to the needs of the

**Table 4. Summary of mechanisms, definitions, and supporting evidence.**

| Name | Definition | Contributing references | Example (quote/ text from manuscript) |
|---|---|---|---|
| **Context** | | | |
| Diagnostic appointment | Appointment when a person receives the official diagnosis of MS within the Neurology services of a hospital. | **Interview ID**: MS_01; MS_04; MS_05; MS_10 HCP_01; HCP_02; HCP_03; HCP_04 EMP_04; EMP_02 **Manuscript:** Jellie (2014); Sweetland (2014); Townsend (2008); Main (2006). | "It's almost at the point of diagnosis that the doctor diagnosing in them, he notices [work problems], contacts another organisation and they step in with this [employment] help to keep or get back in work and how they can help and support you" (MS_01) "Information and support at the time of diagnosis… The consultant neurologist should ensure that people with MS, and with their agreement their family members or carers, are offered oral and written information at the time of diagnosis. This should include, but not be limited to, information about […] legal rights including social care, employment rights and benefits" (NICE, 2022) |
| Routine care | Follow-up appointments when the person with MS receives NHS usual care services to monitor disease progression or presence of new symptoms. | **Interview ID**: EMP_01; EMP_04 MS_02; MS_07; MS_03; MS_04 HCP_02; HCP_03; HCP_04 **Manuscript:** NICE (2022); Sweetland (2014); McGregor (2014); NICE (2014); Sweetland (2010); Townsend (2008); Bisiker & Millinchip (2007); Brewin & Hazell (2004); Kirker (1995) | "What happens if something comes up just after the annual review and have to wait a whole year before you speak something about it… Sometimes changes in ability can be quite subtle" (EMP_04) "The findings indicate that professionals' understanding of the issues affecting the employment of people with MS and the focus of their interventions is dominated by MS and its symptoms. Professionals' awareness of the impact of the personal and larger social environment on an individual's ability to retain employment is less apparent." (Townsend, 2008) |
| **Mechanism** | | | |
| Early intervention | Providing support soon after diagnosis or before a problem arises. In the case of people with MS, this should be at key time points such as at diagnosis, at each yearly review or after a relapse. | **Interview ID**: HCP_01; HCP_03; HCP_02; HCP_04 EMP_01; EMP_03; EMP_04 MS_09; MS_01; MS_02; MS_06; MS_07; MS_08; MS_03; MS_04; MS_05; MS_10 **Manuscript:** Royal College of Physicians (2011); Townsend (2008); Jellie (2014); Sweetland (2014); Main (2006); NICE (2022); McGregor (2014); Kirker (1995); Bisiker & Millinchip (2007). | "I think that a health professional who gives the diagnosis needs to first of all explain what MS is and the context of the MS symptoms, and how it fits into the diagnosis. And then talk about… we can put you in touch with somebody to help you to think through the implications for your work. Just tell us briefly what your work is." (HCP_09) "Early return to work, where possible, may also avoid long periods of inactivity, with loss of confidence and self-esteem." (Main & Haig, 2006, page 290) |
| Individually tailored | Support that matches the specific needs and preferences of a person and their role, and works towards the professional goals and preferences of the person with MS. | **Interview ID**: HCP_02; HCP_03 EMP_03; EMP_04; EMP_02; EMP_05 MS_08; MS_07; MS_06; MS_03; MS_04; MS_09; MS_01; MS_05; MS_10 **Manuscript:** Royal College of Physicians (2011); Townsend (2008); Jellie (2014); Sweetland (2014); Sweetland (2010); Main (2006); NICE (2022); McGregor (2014); NICE (2014); Kirker (1995); Bisiker & Millinchip (2007); Brewin & Hazell (2004). | "You know the triage. If it wasn't there, who decides whether it's a Tier 3 or tier 2, how do you cost that? are you supposed to just have these specialists hanging around waiting for the bat signal?" (EMP_04) "The capacity building aimed to address these barriers and enable health professionals to deliver Tier 1 and Tier 2 of the VR service model. Any issues that could not be addressed by health professionals would be referred to the pilot's case management service, or Tier 3." (McGregor, 2014) |
| Crossing health and employment boundaries | Refers to the interaction between the professionals working in the healthcare setting and other relevant stakeholders from the workplace of the person with MS such as line managers, human resources, occupational health, and co-workers. | **Interview ID**: HCP_01; HCP_02; HCP_03 EMP_02; EMP_03; EMP_01 MS_02; MS_06; MS_08; MS_07; MS_03; MS_04; MS_09; MS_05; MS_10 **Manuscript:** Jellie (2014); Sweetland (2014); Main (2006); Bisiker & Millinchip (2007); Brewin & Hazell (2004). | "I think I would have liked to have a formal document as well to give to my employer of this is what MS is, this is what my MS Nurse has discussed with me, this is what I need, and this is how it affects me." (MS_05) "The guidelines were based on a thorough and comprehensive assessment of a client's abilities and the sharing of information between agencies, the client and the employer. In order to ensure that there were no misunderstandings, recommendations and an action plan would be agreed and documented." (Brewin & Hazell, 2004) |

*(Continued)*

**Table 4.** (Continued)

| Name | Definition | Contributing references | Example (quote/ text from manuscript) |
|---|---|---|---|
| Coordinated effort | Refers to the collaboration between the VR therapist and person with MS (sometimes other key stakeholders are included: employers, disability advisors, and national charities) to organise resources and activities to achieve a desired outcome (i.e., job retention or return to work). | **Interview ID**: HCP_01; HCP_02; HCP_03; HCP_04 EMP_01; EMP_03; EMP_04; EMP_02; EMP_05 MS_01; MS_02; MS_06; MS_07; MS_08; MS_03; MS_04; MS_05; MS_09; MS_10 **Manuscript:** Royal College of Physicians (2011); Townsend (2008); Jellie (2014); Sweetland (2014); Sweetland (2010); Main (2006); McGregor (2014); Kirker (1995); Bisiker & Millinchip (2007); Brewin & Hazell (2004). | "It's just about efficient and effective communication and keeping that really broad range of services all up to date on what's happening with each other... But I guess that would be the biggest barrier in if you didn't have someone brilliant coordinating all [the VR service] might be a bit tricky." (MS_06) "About half of all GPs either could not access specialist vocational rehabilitation at all or they did not know." (Royal College of Physicians, 2011) |
| Empowerment | Process by which people with MS become aware of how their MS symptoms interact with their workplace performance and are able to express their needs and implement strategies to minimise the impact of MS at work. | **Interview ID**: HCP_02 EMP_01 MS_02; MS_06; MS_04 **Manuscript:** Ford (2020); Jellie (2014); Sweetland (2014); McGregor (2014); Sweetland (2010). | "It can really be very uplifting for them [people with MS] to know that they've got all that support [with employment] and it could empower them." (MS_04) "One of the aims of a VR service should be to empower the individual often through education and support." (Sweetland, 2010) |
| **Outcomes** | | | |
| Fostering hope | People with MS acknowledge that MS may cause difficulties at work, but actively work towards approaches to maintaining a positive outlook on the future, knowing there is a team to support them if needed. | **Interview ID**: HCP_03; HCP_04 MS_01; MS_07; MS_10 EMP_04 **Manuscript:** McGregor (2014); NICE (2014); Sweetland (2010); Kirker (1995) | "If you knew you'd got someone [in the NHS] there that you could turn to and actually advise you properly, you would be a lot…you would have to feel a lot more confident going forward." (MS_01) "Initially participants presented with anxiety about their performance at work. Specific worries included job security, job performance both now and in the future" (Sweetland, 2010) |
| Improved workplace relationships | To enhance the quality of the interactions between employer and employee (and sometimes co-workers). The employer and employee develop an understanding of workplace challenges and seek for solutions (which can include reasonable adjustments and agreements) through communication and building trust. | **Interview ID**: HCP_02; HCP_03 MS_06 EMP_01; EMP_04 **Manuscript:** Jellie (2014); Sweetland (2010); Townsend (2008); Bisiker & Millinchip (2007); Main (2006); Brewin & Hazell (2004). | "One of the main indicators in terms of like whether o if return to work, it's going to be successful is the attitude of the employer." (HCP_02) "Through the processes of the intervention participants described they felt enabled to ultimately manage work and workplace relationships more effectively." (Jellie et al. 2014) |
| Reasonable adjustments | Key stakeholders (employer, employee, OH, etc.) agree on providing reasonable adjustments (e.g., physical modifications to environment, flexible working patterns, assistive technology, changing policies, etc.) to minimise the impact of MS at work. | **Interview ID:** HCP_02; HCP_03 MS_07; MS_05 EMP_01; EMP_02 **Manuscript:** McGregor (2014); Jellie (2014); Sweetland (2014); NICE (2014); Sweetland (2010); Bisiker & Millinchip (2007); Main (2006); Brewin & Hazell (2004). | "If a person has been in a job for a longer period of time and has a relationship with their line manager… I think it has much more positive outcomes…the NHS is a prime example, where there might be HR wording where they'll say it's [reasonable adjustments] at the line managers discretion." (HCP_03) "Annabelle's legal rights and options around disclosure at work were discussed. It was agreed that Annabelle would disclose to her HR department and then following their recommendations, decide when to tell her line manager [...] Annabelle recognised that she was protected by the law and had the right to ask for reasonable adjustments at work." (Sweetland et al. 2014) |
| Improved health outcomes | Overall improvement of the health and well-being of the person with MS, driven by the reduction of anxieties regarding their future at work, having meaningful social workplace relationships, and having an improved economic situation. | **Interview ID:** HCP_01; HCP_03 MS_06; MS_07; MS_08; MS_04 EMP_04 **Manuscript:** McGregor (2014); Jellie (2014); Sweetland (2010); Townsend (2008); Bisiker & Millinchip (2007); Kirker (1995) | "it doesn't matter whether your work is a paid job or a volunteer job… doing something and getting up every day and moving around is better for your mobility, for your balance. I think it affects finances. If you give up work, it affects you socially, it affects your mental health." (MS_08) "The evaluation showed that the service was associated with a range of outcomes including improvements in health […] It should be remembered that these clients have severe and enduring health conditions, and any positive shift is important." (McGregor, 2014) |

*(Continued)*

**Table 4.** (Continued)

| Name | Definition | Contributing references | Example (quote/ text from manuscript) |
|---|---|---|---|
| Job retention | People with MS returns to or remains in employment following the intervention. For some, this may involve reducing working hours or changing industry to maximise the number of years they can remain in paid employment. | **Interview ID**: MS_06; MS_07; MS_04 HCP_02; HCP_04 **Manuscript:** Ford (2020); Jellie (2014); Sweetland (2014); McGregor (2014); Sweetland (2010); Bisiker & Millinchip (2007); Main (2006). | "If people get out of the workplace, it's harder to then come back. So, if you support people at the start to keep them in employment rather than them stopping and then supporting them to come back again when…it's going to be better." (MS_07) "Eighteen out of the 25 clients who returned to work had managed to return to their existing jobs, while 7 had found employment in a completely new field. A comparison between diagnoses was also made to see if clients with a particular diagnosis were more successful in returning to work. Twenty-four clients were involved in various forms, and were at various stages, of training. The courses ranged from a few weeks to 1-2 years. No significant patterns emerged. The success of returning to their existing job seemed dependent on a combination of their work-related problems, the job, and the employer." (Bisiker & Millinchip, 2007) |

MS, multiple sclerosis; NICE, National Institute for Health and Care Excellence; EMP, employer; HCP, Healthcare professional; VR, vocational rehabilitation; NICE, National Institute for Health and Care Excellence; OH, occupational health; NHS, National health service; HR, human resources.

person with MS. Key professionals involved in identifying those in need and provision of support included OTs, MS Nurses, case managers, VR specialists, and healthcare professionals with expertise in MS [31,38,40,41,43–46]. The MS Nurse or an OT (if available) can provide information for the lower levels of VR support for those people who need signposting to organisations [15,31,38,41].

Interview participants highlighted that if interventions are individualised to patient needs, then they are more likely to identify the key challenges that the person experiences at work and could support job retention better than generic interventions. Participants also suggested that a flexible intervention delivery (e.g., in person, online, by telephone) could improve intervention adherence, and having different intervention levels could aid resource utilisation (i.e., reduce intervention costs) because not everyone will need the most intensive and resource-intensive intervention levels, leading to a reduced intervention cost. Remaining at work was also reported to contribute to having fewer healthcare appointments because the person experiences an overall improved well-being.

## Crossing health and employment boundaries

If employers (e.g., human resources, line manager, occupational health) engage in discussions with healthcare professionals or receive a letter explaining the needs of the employee with MS at work, then employers could become more confident speaking about disability and work and be able to make an informed decision about what support to provide to the employee with MS. Several studies included a component of employer engagement as an essential intervention component, co-workers were also involved in some instances [31,38–40,42,43,45,46]. Sometimes, employer engagement can involve healthcare professionals writing a report for the employer with recommendations and a plan of action; however, due to fear of discrimination and poor coordination between healthcare and employment services, these reports are not always shared with employers [31,40].

Other common barriers to employer engagement are a lack of knowledge about employment services available for further support, limited funding, or time from healthcare professionals [39,40,42,47].

If employers are informed about examples of reasonable adjustments for their employees with MS, then they will have an opportunity to understand MS and their legal responsibilities better, leading to improved workplace relationships with the employees with MS [38,40,42,46]. Interviews highlighted that if employees with MS receive VR, employers will see a reduction in organisational costs from hiring and training new employees, and their employees will be more satisfied and productive at work.

All interview participants suggested that before the healthcare professional engages with their employer, the person with MS should first consent to this interaction. Employers and participants with MS also questioned the feasibility of employers receiving information from the NHS on their employees' health and employment needs since NHS services are currently overstretched.

### Coordinated effort

Closely related to the previous mechanism, evidence suggests that VR is complex and should be delivered by a range of professionals (e.g., MS nurse, physiotherapist, psychologist, OT) according to the needs of the person with MS at work [38,40,42,43].

If NHS teams can identify and assess the employment needs of people with MS; then, they can refer the person to professionals that can provide advice to address their needs. People with MS will need access to a range of services within the VR service, including medication reviews, physical, and cognitive rehabilitation support [31,38,39,43,45–47]. Over time, the VR service will become more effective at referring people with MS to other services, as the team develops relationships and contacts with different services and organisations [43].

Interview participants reported how the structure of current services does not allow for ongoing monitoring of MS progression and employment needs. Thus, hampering the provision of timely support. Interview participants also suggested the need for the NHS to collaborate with local community services and MS charities to facilitate the service provision and reduce NHS pressure. This coordination between third-sector organisations and healthcare can maximise the support people with MS receive beyond the healthcare setting.

### Empowerment

Closely related to the early intervention mechanism, if people with MS are aware of a service that can offer support with employment before they experience difficulties at work, then they are more likely to request support with employment in a timely manner to prepare for disclosure and regain control of their employment circumstances. This can also improve their feelings of hope about their future at work, reduce anxieties, and eventually reduce the risk of job loss [31,43,45,46].

### Programme theory for VR in the NHS for MS

A logic model of the programme theory was produced based on the interaction between the CMOs identified during the realist evaluation (Fig 2).

This programme theory illustrates the prerequisites needed to integrate VR within the NHS and the pathway people with MS would follow during the intervention. The programme theory also depicts how the service can become more efficient over time by developing networks with NHS and external partners. While the CMO configurations are linked, when included in the logic model, the linear relationships have been removed due to the complexity of the interaction between the different components of the programme theory at various time points.

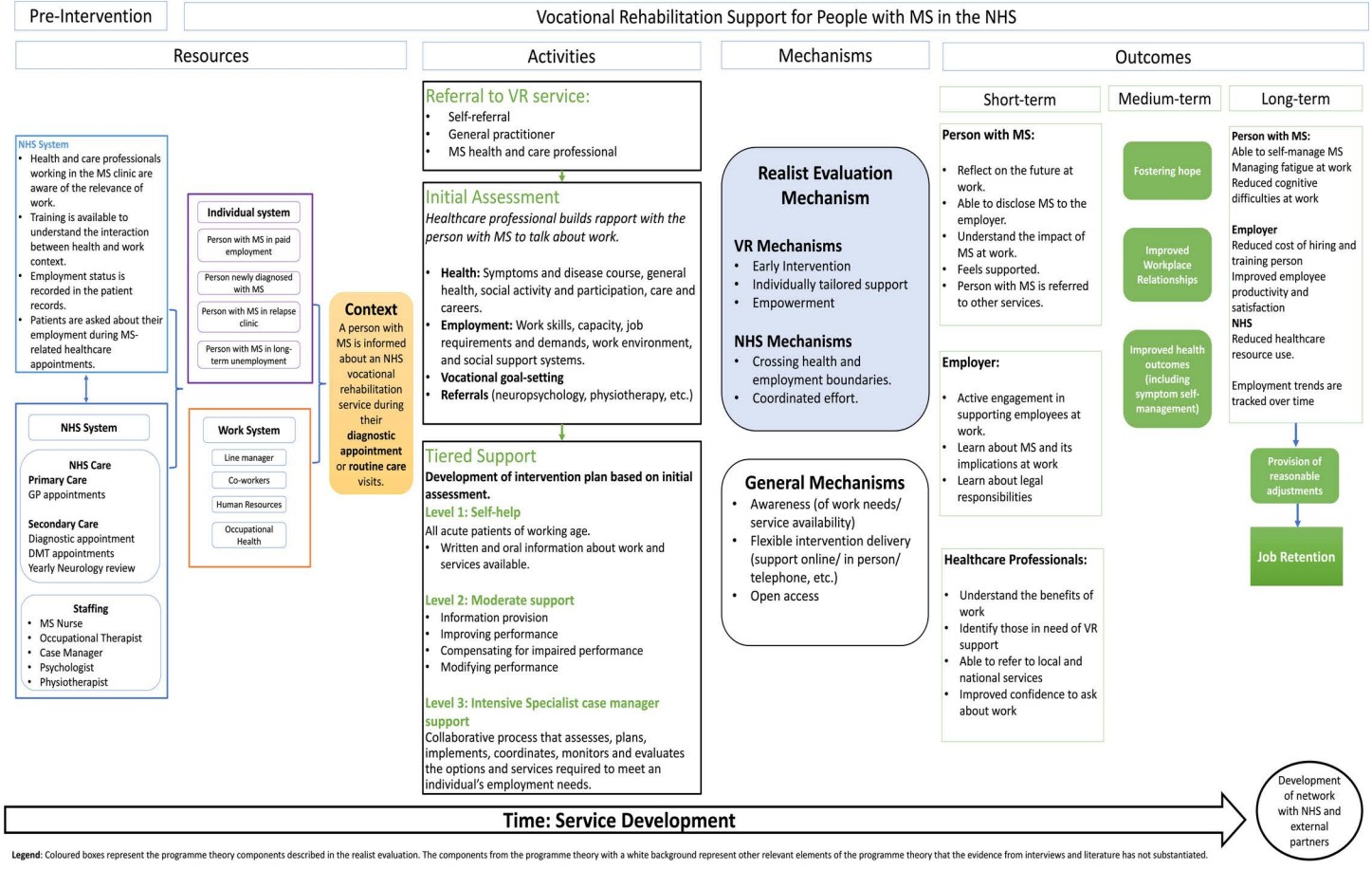

**Fig 2. Logic model of programme theory.**

The mechanisms identified in the realist review have been divided into VR and NHS mechanisms. A section on general mechanisms was included to illustrate mechanisms known to impact VR outcomes that were not identified in the review or interviews but arose from previous studies completed by the authorship team [22,24,49].

The outcomes identified in the review are presented in coloured boxes in the logic model, according to the timeline needed to achieve the outcomes. Job retention is a long-term outcome presented in the programme theory. It is considered a long-term outcome achieved when the person has received VR, learned to self-manage their MS, and the workplace has been adapted to their needs. Other relevant outcomes identified in the interviews and the literature have also been included to represent the broader impact of VR.

## Discussion

This evaluation developed a programme theory about how VR for people with MS could work in the NHS. When VR is provided by the NHS in either a diagnostic or routine appointment, then the early intervention, individualised, empowerment, coordinated effort, cross-boundary approach may lead to job retention through several short-term outcomes. The initial rough programme theory was tested to develop a more robust programme theory for VR in the NHS for MS.

Multiple challenges are associated with implementing employment services for people with MS within existing NHS services. The programme theory identified the need to provide these services through secondary care and MS-specific appointments, as opposed to primary care services (e.g., GPs). This was seen as the most suitable context to identify people in need of support with employment because patients sometimes discuss in MS-specific NHS appointments the impact of their symptoms at work and tend to go to GPs for non-MS-related needs (e.g., colds, back pain).

The evaluation suggested a need for an initial assessment to determine readiness to return to work or risk of job loss [31,38,40,41,43,45,46] and tiered support where people receive support according to the needs reported or identified in the assessment [31,38,41,43,45].

It is well known that there is usually resistance to transforming the culture and organisational structure of such large organisations, like the NHS, due to their complexity [50]. However, we have seen examples of successful changes in MS care, for example, with the development of pathways to provide disease-modifying treatments to people with MS [51]. The success of integrating these new management programmes is primarily due to collaboration between stakeholders and healthcare professional training [51]. In particular, MS specialist nurses are key professionals who should be upskilled to identify employment difficulties or concerns; especially, considering that they are the first point of contact for approximately 90% of people with MS [52].

A key finding of the review relates to the knowledge and expertise of the MS specialist healthcare professionals. Healthcare professionals often do not feel confident discussing employment [24,47]. The review shed light on the fact that although people with MS may ask for employment support at work, their needs are only partially assessed in the NHS because they primarily receive support managing symptoms as opposed to employment-specific support (i.e., worksite visits, managing cognitive problems at work, etc.). This means there is a need to upskill healthcare professionals to identify those needing support and provide them with information or refer them to local services (e.g., national MS charities). A study assessing the primary care-based needs of people with MS failed to identify "employment" as a need of people with MS; however, this could be in part due to the small proportion of participants recruited who were in employment (approximately 25%) [53].

## Clinical and research implications

This evaluation has identified a need to develop training packages for healthcare professionals such as GPs and MS nurses to recognise who may be at risk of job loss following an MS diagnosis. This preventative approach could improve the number of people who receive advice about employment in a timely manner and also lead to people with MS receiving comprehensive care aligned with the NICE guidelines that suggest people with MS should receive advice about employment issues [15].

Clinicians need to include VR within their diagnostic and routine appointments with people with MS. Recording employment status in clinical records can offer information on employment changes following diagnosis, allowing understanding of when post-diagnosis people are more likely to leave the workforce.

Future research should explore the implementation of VR within NHS services to identify organisational, structural, financial, and cultural factors that hamper VR services implementation into routine NHS care. There is also a need to explore how the different components of the programme theory that were not substantiated in this review impact VR outcomes. Future research should also consider exploring NHS policies regarding the funding and provision of VR services for people with MS. Further longitudinal studies are needed to ascertain the long-term impact of VR on healthcare outcomes and resource utilisation.

## Strengths and limitations

A strength of this realist evaluation is our inclusion of people with MS, employers, and healthcare professionals to refine the programme theories developed during the realist evaluation. These key stakeholders support people with health conditions in remaining at work and offer complementary views to refine the theories. This is particularly important, considering the limited literature available on the topic.

The limited number of studies identified in the review, the variability in study design, and the richness of the data extracted are limitations of the study. However, the limited data available in the literature was further substantiated by the semi-structured interviews, which provided rich and detailed information that expanded the initial rough programme theory. Another potential limitation is that some components of the refined programme theory were extracted from the literature or the interviews but were not substantial enough to form full CMOs. Therefore, further research is needed to provide evidence of these aspects. One author conducted the data extraction and synthesis for the review, limiting the rigour of the data analysis. However, the material was reviewed by all manuscript authors at different evaluation phases.

## Conclusion

There is a need to identify people with MS employed at the point of diagnosis to inform them about employment services available. Early intervention can only be possible if healthcare professionals are adequately trained and upskilled to have conversations about work that help people with MS ponder their workplace relationships and ability to work. Current NHS services need restructuring to allow for further routine discussions on changes in symptoms due to the unpredictable nature of MS.

Employment needs to be seen as a healthcare issue to attract further funding and drive the collaboration between employment and health services to achieve a sustainable integration of VR support within existing NHS services for people with MS.

## Supporting Information

**S1 File. RAMESES II reporting standards for realist evaluations: checklist.**
(DOCX)

**S2 File. Search strategy.**
(DOCX)

**S3 File. Data extraction form and quality assessment.**
(DOCX)

**S4 File. Criteria for scoring Rigor and Relevance.**
(DOCX)

**S5 File. Interview topic guides.** Semi-structured interview topic guide.
(DOCX)

**S6 File. Quality appraisal.** S1 Table Quality appraisal summary.
(DOCX)

## Acknowledgements

We thank the research participants for their time and involvement in the research.

## Author contributions

**Conceptualization:** Blanca De Dios Pérez, Vicky Booth, Roshan das Nair, Nikos Evangelou, Juliet Hassard, Helen L. Ford, Ian Newsome, Kate Radford.

**Data curation:** Blanca De Dios Pérez, Vicky Booth, Kate Radford.

**Formal analysis:** Blanca De Dios Pérez, Vicky Booth, Kate Radford.

**Funding acquisition:** Blanca De Dios Pérez, Vicky Booth, Roshan das Nair, Nikos Evangelou, Juliet Hassard, Helen L. Ford, Ian Newsome, Kate Radford.

**Investigation:** Blanca De Dios Pérez, Roshan das Nair, Nikos Evangelou, Juliet Hassard, Helen L. Ford, Ian Newsome, Kate Radford.

**Methodology:** Blanca De Dios Pérez, Vicky Booth, Roshan das Nair, Nikos Evangelou, Juliet Hassard, Helen L. Ford, Ian Newsome, Kate Radford.

**Project administration:** Blanca De Dios Pérez, Vicky Booth, Kate Radford.

**Resources:** Blanca De Dios Pérez, Kate Radford.

**Supervision:** Vicky Booth, Roshan das Nair, Nikos Evangelou, Juliet Hassard, Helen L. Ford, Kate Radford.

**Writing – original draft:** Blanca De Dios Pérez, Vicky Booth, Kate Radford.

**Writing – review & editing:** Vicky Booth, Roshan das Nair, Nikos Evangelou, Juliet Hassard, Helen L. Ford, Ian Newsome, Kate Radford.

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
