## [Decision Letter · Decision Letter 0]

16 Oct 2024

PONE-D-24-16183Vocational rehabilitation for people with multiple sclerosis in the national health service of the United Kingdom: A realist evaluationPLOS ONE

Dear Dr. De Dios Pérez,

Thank you for submitting your manuscript to PLOS ONE. After careful consideration, we feel that it has merit but does not fully meet PLOS ONE’s publication criteria as it currently stands. Therefore, we invite you to submit a revised version of the manuscript that addresses the points raised during the review process.

We look forward to receiving your revised manuscript.

Kind regards,

Diphale Joyce Mothabeng, PhD

Academic Editor

PLOS ONE

Journal Requirements:

The UK Multiple Sclerosis Society funded this research through a research project grant (award:140).  

3. "In the online submission form, you indicated that data will be made available upon reasonable request to the corresponding author due to ongoing data analysis for further research. 

4. Please include a caption for figure 2.

5. We note you have included a table to which you do not refer in the text of your manuscript. Please ensure that you refer to Table 2 in your text; if accepted, production will need this reference to link the reader to the Table.

Additional Editor Comments :

Thank you for your submission. Kindly attend to the reviewers comments.

Reviewers' comments:

Reviewer's Responses to Questions

**Comments to the Author**

1. Is the manuscript technically sound, and do the data support the conclusions?

Reviewer #1: Yes

Reviewer #2: Yes

2. Has the statistical analysis been performed appropriately and rigorously? 

Reviewer #1: Yes

Reviewer #2: Yes

3. Have the authors made all data underlying the findings in their manuscript fully available?

Reviewer #1: Yes

Reviewer #2: Yes

4. Is the manuscript presented in an intelligible fashion and written in standard English?

Reviewer #1: Yes

Reviewer #2: Yes

5. Review Comments to the Author

Reviewer #1: 1. Available literature and as substantiated by the semi-structured interviews support the conclusion. However, can the authors considered the following

43 - being specific as to what the resulting program theory

45 - add 'support and empower the employee'.

47 - add - for example job retention

48 -49: With reference to NHS there are many changes that can be suggested - is it possible briefly outline the change required for this patient group

148 - I wondered what was the structure of the interview schedule perhaps clarify

169- Data analysis - the steps were clearly outlines - Is it possible to comment on the method for the interviews e.g. was it Thematic analysis - inductive/deductive etc.

176 - what were the resultant codes

226 - please clarify who is the public in the PPP

Overall comments:

Easy to read and professional presented paper relevant literature was included and COM helped to structure the the delivery of information. The study highlighted other findings that might be contributing to the failure of service delivery e.g. professional skills and fragmented services as well as the need of the healthcare professional to also be educated on the legal framework regarding employee retention in the workplace.

Data Management - I would like to suggest to the authors to outline more e.g. where they are keeping the transcribed data and for how long and the security around that.

Reviewer #2: Overview

First, I commend the authors for the efforts and rigour invested into this study. This realistic evaluation sought to understand what VR interventions for people with MS delivered in the NHS of the UK work, under what circumstances, and why. The resulting programme theory illustrates a complex interplay between the healthcare and employment sectors to influence health and employment outcomes. VR programmes that offer timely support, tailored to the needs of the person with MS, and that support the employee beyond the healthcare context are most likely associated with improved employment outcomes. However, a number of revisions are required to put the manuscript in its best state.

Areas of improvements

The major areas of improvement include rearranging the objectives, the rough programme theory, streaming the definition of context to reflect the same concept in both the review and the interview arm and limitation. A few minor revisions are highlighted in the manuscript text.

Objectives

I would prefer that the specific objectives be re-arranged to ensure a systematism in their flow. I would think that “what works/context (VR interventions/resources and opportunities) should come first, then then the outcome and the mechanism that produced the outcome. That way, it remains in tune with CMO configuration. The present arrangement/flow is difficult to follow.

Rough programme theory

Whereas I acknowledge that the cited study [22] covers some of the facts outline in the rough programme theory, it does not do so completely. Hence, I would think that other relevant studies namely 1. doi: 10.3233/WOR-210012. 2. doi: 10.1177/1352458507078414 which speak into the fact of the rough programme theory should be cited. A theory is a collection of ideas/principles upon a practice/intervention may be based. Citing only self when there are other studies that speak into the fact of the rough programme theory makes the proposed theory weak. Besides, the more consistent the facts of a theory, the higher the application, utilization and universal transformation into a law/policy. In addition, I would prefer that the authors cite each of the facts presented in the rough programme. This will allow for confirmation of such facts in the cited studies. Also, it is difficult to tell which facts of the proposed theory is/could be measured at micro-level, meso and/or macro level. This should be clarified.

Data Analysis

This definition does not align with the context as highlighted in the rough programme theory. Here context referred to setting or stage. How was it different from the use of context in the rough programme theory? I think it would make sense to define the CMO and operationalize at their first appearance. Definition should be as consistent as possible in both the reviews and the qualitative arm/interviews.

Limitations

How does the variability in study characteristic contributes to the limitations?

Verdict

I consider the data provided in this manuscript essential given their relevance to subject of study. However, the manuscript requires some revisions to coherence and improve readability.

6. PLOS authors have the option to publish the peer review history of their article (what does this mean? ). If published, this will include your full peer review and any attached files.

**Do you want your identity to be public for this peer review?** For information about this choice, including consent withdrawal, please see our Privacy Policy .

Reviewer #1: No

Reviewer #2: No

---

## [Author Response · Author response to Decision Letter 1]

2 Jan 2025

We would like to thank the editor and reviewers for their time reading the manuscript and offering suggestions to improve our work.

We have reviewed the manuscript according to the PLOS ONE style requirements and made the required changes.

2. Thank you for stating the following financial disclosure: The UK Multiple Sclerosis Society funded this research through a research project grant (award:140). Please state what role the funders took in the study. If the funders had no role, please state: ""The funders had no role in study design, data collection and analysis, decision to publish, or preparation of the manuscript."" If this statement is not correct you must amend it as needed. Please include this amended Role of Funder statement in your cover letter; we will change the online submission form on your behalf.

Thank you for this comment. The funder had no role in the study. Therefore, the role of the funder statement should be updated to read: “The funders had no role in study design, data collection and analysis, decision to publish, or preparation of the manuscript."

3. "In the online submission form, you indicated that data will be made available upon reasonable request to the corresponding author due to ongoing data analysis for further research. All PLOS journals now require all data underlying the findings described in their manuscript to be freely available to other researchers, either 1. In a public repository, 2. Within the manuscript itself, or 3. Uploaded as supplementary information. This policy applies to all data except where public deposition would breach compliance with the protocol approved by your research ethics board. If your data cannot be made publicly available for ethical or legal reasons (e.g., public availability would compromise patient privacy), please explain your reasons on resubmission and your exemption request will be escalated for approval.

Unfortunately, we cannot make the interview transcripts available because doing so would breach the ethical approval received from the University of Nottingham and Health Research Authority (HRA). In the participant information sheet (PIS) and consent form, participants were informed that only quotes from their interviews would be used for publication and presentations at conferences or other events. However, they were not informed that the interview transcripts would be made publicly available. We are able to provide the data extraction forms for the studies included in the realist evaluation.

Therefore, we have updated our statement to:

There are restrictions related to the consent provided by the study participants for access to the data they provided for the study. The University of Nottingham (UoN), the Health Research Authority (HRA) for the United Kingdom (UK), and the London Stanmore Research Ethics Committee (REC) approved the study.

The University of Nottingham is legally responsible for data security, and the Chief Investigator (Prof. Kate Radford) is the Data Custodian managing the access to the data. Reasonable request for access to the data reported in this study may be sent to the corresponding author blanca.dediosperez@nottingham.ac.uk

The data extraction forms from the Realist Evaluation have been included in a Data Repository and can be accessed through the following link: https://rdmc.nottingham.ac.uk/handle/internal/11597

4. Please include a caption for figure 2.

Figure 2 already included the caption on the manuscript on page 28 (Fig 2: Logic model of programme theory)

5. We note you have included a table to which you do not refer in the text of your manuscript. Please ensure that you refer to Table 2 in your text; if accepted, production will need this reference to link the reader to the Table.

We apologise for the mistake. We accidentally wrote Table 1 on page 12 instead of mentioning Table 2. We have updated the text on page 12 to mention Table 2, which was the table we wanted to cite there.

The in-text citations have been updated, after we identified one reference that was not correctly cited. The Supporting information was reported correctly in the manuscript. This has been reviewed to make sure it is correct.

We have reviewed the reference list, and we can confirm that now it is correct and complete. There are no retracted articles cited in the manuscript.

Reviewer #1:

8) Available literature and as substantiated by the semi-structured interviews support the conclusion. However, can the authors considered the following

We thank the reviewer for their time reading the manuscript and for providing feedback to improve the content and readability of the manuscript.

9) 43 - being specific as to what the resulting program theory

We have included the following explanation on the abstract: The resulting programme theory explains the implementation of VR in the NHS for MS populations, uncovering the complex interplay between the healthcare and employment sectors to influence health and employment outcomes.

We were unable to write more in the abstract due to the word count.

10) 45 - add 'support and empower the employee'.

We have included in line 45 the text requested by the reviewer.

11) 47 - add - for example job retention

We have included in line 47 the text requested by the reviewer.

12) 48 -49: With reference to NHS there are many changes that can be suggested - is it possible briefly outline the change required for this patient group

We have included the following statement on lines 49 and 51 of the abstract:

Embedding VR support within the NHS requires substantial cultural and organisational change (e.g., increased staff numbers, training, and awareness about the benefits of work).

13) 148 - I wondered what was the structure of the interview schedule perhaps clarify

We included in Supplementary File 4 the interview schedule (i.e., interview topic guide) to offer further information on the interview. This is reported on Line 159, page 8. Therefore, we did not make any further changes to the text.

14) 169- Data analysis - the steps were clearly outlined - Is it possible to comment on the method for the interviews, e.g. was it Thematic analysis - inductive/deductive, etc?

The interviews were analysed following a realist evaluation approach and using the standardised structure of building and refining CMO configurations (https://journals.sagepub.com/doi/pdf/10.1177/1609406919859754). We have updated the manuscript explaining the fact that we followed an inductive approach. The paragraph on page 10 lines 207-211 now reads: “Interviews were audio-recorded, transcribed verbatim, and systematically coded following a hybrid deductive and inductive approach. Data were analysed deductively according to CMO configurations following a realist evaluation approach [35,37], followed by an inductive data-led analysis to develop further our understanding of how VR could work in the NHS.”

15) 176 - what were the resultant codes

Table 4, page 20 presents the resultant codes in the manuscript.

16) 226 - please clarify who is the public in the PPP

We have included the Initials (IN) and demographic information on the PPI representative who supported this study. He is also a named author on the paper. On Line 229 we state: A patient and public involvement (PPI) representative (IN, White British Man) was involved in developing the study protocol, topic guide, and data analysis to improve the validity of the findings and ensure patients’ perspectives are included in the research.

17) Overall comments: Easy to read and professional presented paper relevant literature was included and COM helped to structure the the delivery of information. The study highlighted other findings that might be contributing to the failure of service delivery e.g. professional skills and fragmented services as well as the need of the healthcare professional to also be educated on the legal framework regarding employee retention in the workplace.

We thank the reviewer for their time reading the manuscript and providing input to improve its quality.

18) Data Management - I would like to suggest to the authors to outline more e.g. where they are keeping the transcribed data and for how long and the security around that.

The data will be kept for seven years following the University of Nottingham regulations. The transcripts cannot be made available for ethical reasons (i.e., participants did not consent to the full transcript being publicly available) as explained in comment #3. However, we have now made available on a public repository data extracted from the studies. This can be found on the following link: https://rdmc.nottingham.ac.uk/handle/internal/11597

Reviewer 2

Overview

19) First, I commend the authors for the efforts and rigour invested into this study. This realistic evaluation sought to understand what VR interventions for people with MS delivered in the NHS of the UK work, under what circumstances, and why. The resulting programme theory illustrates a complex interplay between the healthcare and employment sectors to influence health and employment outcomes. VR programmes that offer timely support, tailored to the needs of the person with MS, and that support the employee beyond the healthcare context are most likely associated with improved employment outcomes. However, a number of revisions are required to put the manuscript in its best state.

We thank the reviewer for taking the time to review the manuscript and for the detailed comments that helped us improve the way we conveyed the message of this study. We have addressed the reviewer’s comments and responded below.

Objectives

20) I would prefer that the specific objectives be re-arranged to ensure a systematism in their flow. I would think that “what works/context (VR interventions/resources and opportunities) should come first, then then the outcome and the mechanism that produced the outcome. That way, it remains in tune with CMO configuration. The present arrangement/flow is difficult to follow.

We thank the reviewer for this comment. We have rearranged the objectives to ensure the follow the CMO configuration. This change has been included on page 6. With the new order, the objects are presented as follows:

The realist review will answer the following questions: (1) What are the important contexts within the NHS that determine whether the different mechanisms within a VR intervention for employed people with MS produce the intended outcomes?; (2) What are the mechanisms, acting at an individual and organisational level, by which VR interventions for employed people with MS produce the intended outcomes (e.g., job retention)?; (3) What are the possible outcomes of a VR intervention for employed people with MS?

Rough programme theory

21) Whereas I acknowledge that the cited study [22] covers some of the facts outline in the rough programme theory, it does not do so completely. Hence, I would think that other relevant studies namely 1. doi: 10.3233/WOR-210012. 2. doi: 10.1177/1352458507078414 which speak into the fact of the rough programme theory should be cited. A theory is a collection of ideas/principles upon a practice/intervention may be based. Citing only self when there are other studies that speak into the fact of the rough programme theory makes the proposed theory weak. Besides, the more consistent the facts of a theory, the higher the application, utilization and universal transformation into a law/policy. In addition, I would prefer that the authors cite each of the facts presented in the rough programme. This will allow for confirmation of such facts in the cited studies. Also, it is difficult to tell which facts of the proposed theory is/could be measured at micro-level, meso and/or macro level. This should be clarified.

We thank the reviewer for this comment. We have now included the references for the suggested studies (De Dios Perez et al. 2022 and Sweetland et al. 2007). We have also included a statement reflecting that the development of the initial programme theory was also through discussion with the manuscript's authors.

Our group has conducted extensive work in vocational rehabilitation (VR) for people with MS, which led to the development of a job retention VR intervention following the person-based approach (https://doi.org/10.1177/02692155241235956), which helped us identify the key aspects of VR to inform our theory. The Initial Rough Programme Theory has been developed over the years based on our understanding of VR in MS. Therefore some components of the Initial Programme Theory were discussed as relevant by the authors due to their experience and some informed by literature.

We have updated Table 1 to include information on the evidence underpinning each section of the initial programme theory and to represent what aspects can be measured at the micro, meso, and macro levels. For further information on the evidence underpinning the final components of the revised programme theory, Table 4 includes information on which references from the literature on the topic (external to the team) contributed to each component of the programme theory.

Data Analysis

22) This definition does not align with the context as highlighted in the rough programme theory. Here context referred to setting or stage. How was it different from the use of context in the rough programme theory? I think it would make sense to define the CMO and operationalize at their first appearance. Definition should be as consistent as possible in both the reviews and the qualitative arm/interviews.

The information about CMO and its definition has been moved to the initial programme theory section of the manuscript (page 7) to facilitate understanding the initial programme theory. During the initial rough programme theory development, the authors were unclear about the context where the intervention would fit. Therefore, the initial theory includes a “vague” context description (i.e., usual care). We were interested in any healthcare-related context, and this allowed us to search the literature with a broader range of contexts in mind.

In the resulting programme theory, our final contexts were (1) the diagnostic appointment and (2) routine MS appointments, which routinely occur in the care of people with MS.

Limitations

23) How does the variability in study characteristic contributes to the limitations?

The section on the limitations of the realist evaluation provides insight into the fact that a key limitation is the limited number of studies and the “richness” of the data extracted. In Realist research, all study designs can be included to offer insight and information about how interventions work, for whom, and under what circumstances. Therefore, the focus is on how rich the data are (assessed during the quality appraisal for rigour and relevance of the studies to the programme theory (reported on page 16 and S6 Table), as opposed to the study design.

However, based

---

## [Decision Letter · Decision Letter 1]

30 Jan 2025

Vocational rehabilitation for people with multiple sclerosis in the national health service of the United Kingdom: A realist evaluation

PONE-D-24-16183R1

Dear Dr. De Dios Pérez,

We’re pleased to inform you that your manuscript has been judged scientifically suitable for publication and will be formally accepted for publication once it meets all outstanding technical requirements.

Kind regards,

Omid Beiki, M.D., Ph.D.

Academic Editor

PLOS ONE

Additional Editor Comments (optional):

Reviewers' comments:

Reviewer's Responses to Questions

**Comments to the Author**

1. If the authors have adequately addressed your comments raised in a previous round of review and you feel that this manuscript is now acceptable for publication, you may indicate that here to bypass the “Comments to the Author” section, enter your conflict of interest statement in the “Confidential to Editor” section, and submit your "Accept" recommendation.

Reviewer #1: All comments have been addressed

Reviewer #2: All comments have been addressed

2. Is the manuscript technically sound, and do the data support the conclusions?

Reviewer #1: Yes

Reviewer #2: Yes

3. Has the statistical analysis been performed appropriately and rigorously? 

Reviewer #1: Yes

Reviewer #2: Yes

4. Have the authors made all data underlying the findings in their manuscript fully available?

Reviewer #1: Yes

Reviewer #2: Yes

5. Is the manuscript presented in an intelligible fashion and written in standard English?

Reviewer #1: Yes

Reviewer #2: Yes

6. Review Comments to the Author

Reviewer #1: I commend the authors for their efforts in addressing the reviewers' comments. The paper is well-written, and the results, outcomes, and value-added contributions are clearly articulated.

Reviewer #2: First, I commend the authors for the efforts and rigour invested into this study. This realistic evaluation sought to understand what VR interventions for people with MS delivered in the NHS of the UK work, under what circumstances, and why. The resulting programme theory illustrates a complex interplay between the healthcare and employment sectors to influence health and employment outcomes. VR programmes that offer timely support, tailored to the needs of the person with MS, and that support the employee beyond the healthcare context are most likely associated with improved employment outcomes.

The comments have been responded to, and suggestions applied. The manuscript readability has improved significant and I recommend it for publication in PLOS ONE.

7. PLOS authors have the option to publish the peer review history of their article (what does this mean? ). If published, this will include your full peer review and any attached files.

**Do you want your identity to be public for this peer review?** For information about this choice, including consent withdrawal, please see our Privacy Policy .

Reviewer #1: **Yes: ** Ntombizivumile Hankwebe

Reviewer #2: No

---

## [Editor Report · Acceptance letter]

PONE-D-24-16183R1

PLOS ONE

Dear Dr. De Dios Pérez,

I'm pleased to inform you that your manuscript has been deemed suitable for publication in PLOS ONE. Congratulations! Your manuscript is now being handed over to our production team.

Kind regards,

on behalf of

Dr. Omid Beiki

Academic Editor

PLOS ONE